# Bubble-water/catalyst triphase interface microenvironment accelerates photocatalytic OER via optimizing semi-hydrophobic OH radical

Guanhua Ren [1,3], Min Zhou[1,3], Peijun Hu [1,2], Jian-Fu Chen [1] & Haifeng Wang [1] ✉

Photocatalytic water splitting (PWS) as the holy grail reaction for solar-to-chemical energy conversion is challenged by sluggish oxygen evolution reaction (OER) at water/catalyst interface. Experimental evidence interestingly shows that temperature can significantly accelerate OER, but the atomic-level mechanism remains elusive in both experiment and theory. In contrast to the traditional Arrhenius-type temperature dependence, we quantitatively prove for the first time that the temperature-induced interface microenvironment variation, particularly the formation of bubble-water/TiO$_2$(110) triphase interface, has a drastic influence on optimizing the OER kinetics. We demonstrate that liquid-vapor coexistence state creates a disordered and loose hydrogen-bond network while preserving the proton transfer channel, which greatly facilitates the formation of semi-hydrophobic ˙OH radical and O-O coupling, thereby accelerating OER. Furthermore, we propose that adding a hydrophobic substance onto TiO$_2$(110) can manipulate the local microenvironment to enhance OER without additional thermal energy input. This result could open new possibilities for PWS catalyst design.

Understanding and optimizing the water/catalyst interface, which offers a unique microenvironment for reactions to occur, is fundamentally important and highly attractive in the field of heterogeneous catalysis and others[1-4]. In the vitally important case of photocatalytic water splitting to produce hydrogen and oxygen occurring at aqueous interface (corresponding to hydrogen/oxygen evolution reaction, HER/OER)[5-11], recent studies suggested that the catalytic activity of the kinetically sluggish OER process can interestingly undergo a substantial increase at elevated temperatures[12-16]; specifically, the reaction rate in the boiled aqueous solution generated at high temperature is observably higher than that at the liquid-water/catalyst interface at room temperature[4,17]. This has attracted widespread research interest and generated considerable debate. Generally, the origin of activity enhancement with temperature (T) is ascribed to the accelerated reaction kinetics using the Arrhenius equation, $k = A \times \exp(-E_a/RT)$, in which the rate constant, $k$, is enlarged as T increases, while the other parameters remain constants (defined as the traditional temperature effect hereinafter)[18-20]. However, it was also reported that even with the increased temperature, there is almost no photocatalytic activity in the steam-water phase environment under normal pressure[21], possibly due to the fact that the vapor phase in a macro-/micro-bubble environment is significantly less dense than liquid, and the number of molecules close to the active sites and the reaction probabilities are lower.

[1]State Key Laboratory of Green Chemical Engineering and Industrial Catalysis, Centre for Computational Chemistry and Research Institute of Industrial Catalysis, East China University of Science and Technology, Shanghai 200237, China. [2]School of Chemistry and Chemical Engineering, Queen's University Belfast, Belfast, UK. [3]These authors contributed equally: Guanhua Ren, Min Zhou. ✉e-mail: hfwang@ecust.edu.cn

Overall, it is therefore imperative to understand the complete mechanism underlying these phenomena.

So far, several experiments have provided evidence of enhanced OER activity at elevated temperatures beyond the traditional temperature factor. Nurlaela et al. observed a decrease in the apparent activation energy of OER under visible light irradiation within the temperature range of 275–348 K[19]. Similarly, Li and colleagues utilized a photothermal substrate to convert liquid water into steam-water, revealing significantly higher OER activity at the steam-water/catalyst interface compared to the liquid-water/catalyst interface[4,17]. On the other hand, subsequent investigations have also demonstrated that water splitting can be improved by increasing the water coverage and reaction pressure in the steam-water reaction environment[21,22]. These findings highlight the significance of the liquid/solid interface microenvironment in driving photocatalytic activity enhancement, indicating that the traditional temperature effect alone cannot fully explain the observed activity enhancement. Noticeably, Wang et al. very recently developed an interesting floatable platform that exhibits impressive photocatalytic efficiency by facilitating the reaction at air-water biphase environment[23]. Therefore, it is of paramount importance to understand the chemical role of the interface microenvironment in general. Specifically, it is pivotal to disclose the relationship among the interface microenvironment, reaction temperature, and reaction activity in the photocatalytic process.

Although the microenvironment may be expected to be of importance, to access how significant a role it can play in reaction processes is very challenging. Experimentally, it is extremely demanding to probe into the dynamic behavior of interface structures and reactions at the water/catalyst interface in situ. Theoretically, to model the systems with such complexity, particularly the photocatalytic reaction occurring on the excited semiconductor surface, is equally challenging due to the difficulties in simulating surface radical intermediates, the aqueous environment, and the complex reaction network. Overall, there are still crucial questions that need to be answered explicitly: (i) How does the temperature affect the liquid/solid interface environment? (ii) As compared to the traditional temperature effect, how is the photocatalytic OER activity influenced by the distinct interfaces under elevated temperatures and what is the inherent mechanism for such an activity response? (iii) Is it possible to manipulate the interface environment to boost OER at less thermal energy input (i.e., avoiding high temperature), and if so, what approach can be employed?

Herein, we chose the most widely studied rutile-$TiO_2(110)$ surface[24,25] as the photocatalyst and compared the OER activities at different water/$TiO_2(110)$ interface environments synchronized with different temperatures, utilizing the recently developed multi-point averaging molecular dynamics (MPA-MD)[26] together with the first principles based microkinetic analysis[27]. An unexpected interface environment effect on the performance of OER caused by temperatures was discovered. We proved that the ˙OH radical as the pivotal intermediate of photocatalytic OER exhibits a unique relative hydrophobic feature, and the interface environment change due to temperature change influence the formation of ˙OH radical. We demonstrated that the microenvironment plays a huge role in chemical reactions, and the liquid-vapor coexistence environment induced by elevating temperature is pivotal to improve the OER activity. Moreover, we show that by introducing the hydrophobic organic molecules on the surface to modify the reaction microenvironment at the interface, the efficiency of photocatalytic OER can be improved by a factor of 25 at room temperature. This work represents one of the first attempts to quantitatively determine the water/catalyst interface microenvironment effect at the atomic level, and open up a new avenue for optimizing catalytic performances by manipulating interface microenvironments.

## Results

### Interface environments and OER activities at different temperatures

Figure 1a-f depict the specific water/$TiO_2(110)$ interface environments at different temperatures, showcasing characteristic snapshots from the quasi-equilibrium stage obtained from the AIMD simulations (covering the last ~4 ps duration). The profiles show the average water density distributions as a function of the O-catalyst distance perpendicular to $TiO_2(110)$, and the water density distribution profiles of first two layers are compared in Fig. 1g. Note that in this study, the interface environments at different temperatures are denoted as "temperature (state)"; for example, 298 K (l) represents the liquid-state water at 298 K, and 500 K (coexist) is the coexistence state of vapor and liquid at 500 K. It can be seen that as the temperature increases, the interfacial water network distribution is affected to a different extent. At room temperature (Fig. 1a), water molecules close to $TiO_2(110)$ bind to the $Ti_{5c}$ sites and form the first-layer chemisorbed water, and the second layer of water molecules contact the first water layer. The other water layers above the second layer resemble the bulk aqueous water, with the densities oscillating around the density of bulk water (-1 g/$cm^3$). As the temperature increases before reaching the liquid-gas phase transition (at 373 K (l); Fig. 1b), there is a slight decline in the density of the first layers, while the second layer water shifts up and the density increase slightly. At the 500 K (l) state stabilized at a high pressure (-27.8 atm), the hydrogen bonds are more damaged and the water tends to move upwards, so that the densities of the first and second layers show a decrease compared to that at 373 K (l). For comparison, we simulated the vaporization process of liquid water at 500 K under a high-pressure condition (Fig. 1c–f). As can be seen from Fig. 1d–f, as the external pressure decreases from 500 K (l), the volume of the surface water layer increases and the hydrogen bonding network is further damaged; this results in a dramatic decrease not only in the the density of the first- and second-layer water, but also in the bulk water density. Moreover, there is a clear reduction in the average number of hydrogen bonds per water molecule, accompanied by a noticeable increase in the average length of these hydrogen bonds (Supplementary Fig. 1). Consequently, the micro-bubbles diffuse stochastically in the aqueous region. It is worth emphasizing that as the temperature increases and the water density decreases, the surface coverage of water decreases and the bond length of water adsorption lengthens (see the statistical analysis shown in Supplementary Fig. 2 and Table 1).

To comprehensively disclose the interface effects on the OER performance caused by temperature changes, it is necessary to disentangle and analyze the individual influencing factor, namely the traditional temperature effect and the microenvironment of water/$TiO_2(110)$ interface. Firstly, the enthalpy changes of all the elementary steps in the OER process at different water/$TiO_2(110)$ interfaces and the related reaction barriers were calculated by utilizing the MPA-MD methodology (Supplementary Table 2). Note that the widely accepted reaction mechanism of photocatalytic OER on $TiO_2(110)$ involves several key steps (see reaction scheme in Supplementary Fig. 3)[26]. Initially, the dissociation of the adsorbed water ($H_2O_{ad}$) occurs first via proton transfer (PT), forming a Zundel-like hydrated proton ($H_5O_2^+$) in the water environment and generating terminal hydroxyl $OH_t^-$ on the surface $Ti_{5c}$ site. Next, the photogenerated hole transferred from the bulk $TiO_2$ to the surface can be trapped at $OH_t^-$, leading to the formation of the key intermediate ˙$OH_t$ radical ($OH_t^- + h^+ \rightarrow$ ˙$OH_t$). Further, the ˙$OH_t$ radical undergoes deprotonation and yields another key $O_t^-$ radical on $Ti_{5c}$ site through a similar PT mode (˙$OH_t \rightarrow O_t^- + H^+$). The newly produced $O_t^-$ radical can couple with another adjacent $O_t^-$ to yield $O_2^{2-}$, which can eventually generate $O_2$ by capturing two successive holes ($O_2^{2-} + 2h^+ \rightarrow O_2$).

By utilizing the obtained energetics of each elementary step, we are able to conduct microkinetic analyses to quantitatively evaluate

the OER activity at different interfacial environments. Initially, we estimated the OER rates by employing the steady-state microkinetic model, in which the concentration of surface-reaching holes ($h^+$) was assumed to be ~$10^{-9}$ monolayer (ML)[26]. Figure 1h shows the calculated OER rates at different water/TiO$_2$(110) interfaces as a function of temperature. From 298 K (l) to 373 K (l), the photocatalytic OER activity increases with temperatures as expected, with an increase from 0.01 site$^{-1} \cdot$s$^{-1}$ to 0.43 site$^{-1} \cdot$s$^{-1}$; when the temperature is further increased to 500 K while maintaining the liquid-phase state, the OER rate at 500 K (l) escalates to 7.48 site$^{-1} \cdot$s$^{-1}$. For comparison, if we hypothetically postulate that the interfacial environment remains unchanged at elevated temperature (as at 298 K (l)) to assess the impact of interfacial conditions, the calculated OER rates at 373 K and 500 K (marked as 373 K (l$_{298}$) and 500 K (l$_{298}$) in Fig. 1h, respectively) are only 0.15 site$^{-1} \cdot$s$^{-1}$ and 2.32 site$^{-1} \cdot$s$^{-1}$, respectively; in this situation, there is a linear relationship between the reciprocal of the temperature (1/T) and these logarithmic rates of 298 K (l), 373 K(l$_{298}$) and 500 K (l$_{298}$) (Fig. 1h, black dotted line), as expected from the common Arrhenius equation, $k = A \times \exp(-E_a/RT)$, where

$E_a$ is essentially a constant. In reality, however, it can be seen that there is an intriguing deviation from linearity in the temperature dependence of the OER activity, as indicated by the upward curvature of the red curve in Fig. 1h. These results reveal that the activation energies are altered with increasing temperature due to the influence of interfacial environment. Remarkably, when the liquid-water partially vaporizes and a vapor-liquid coexistence state (referred to as the 500 K (coexist) condition) is formed, the OER activity undergoes a dramatic improvement and reaches 38.36 site$^{-1} \cdot$s$^{-1}$. This value is approximately 16.5 times higher than that observed at the same temperature without interfacial changes (2.32 site$^{-1} \cdot$s$^{-1}$). It is worth noting that we also performed an investigation into the influence of the concentration surface-reaching hole on the OER rate, covering a range from $10^{-10}$ to 1 ML (Supplementary Fig. 4). Remarkably, the system consistently exhibits excellent OER activity under the 500 K (coexist) condition, which aligns with the activity trend described in Fig. 1h. These findings highlight the importance of interfacial microenvironments, in addition to temperature, in influencing the reaction rates.

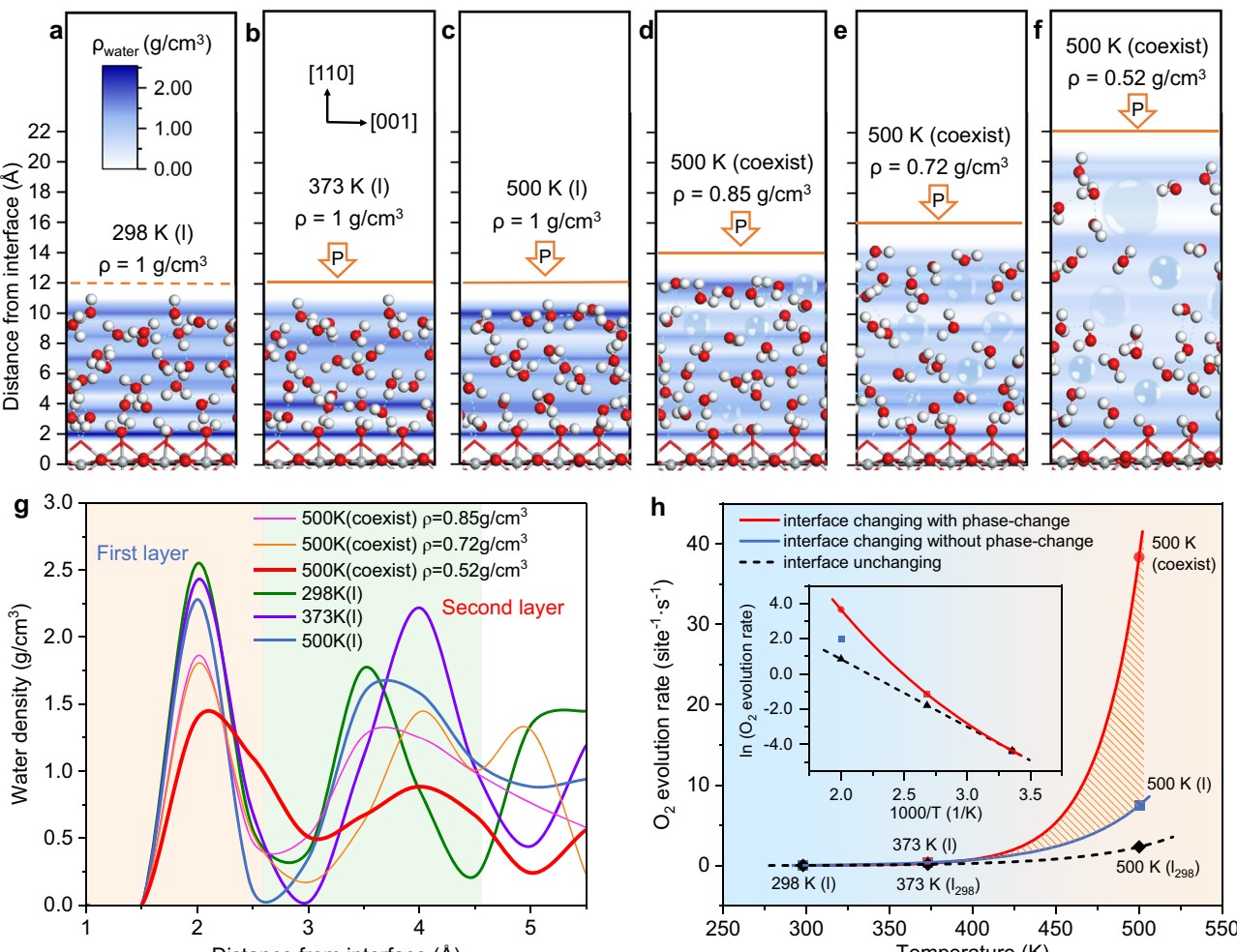

**Fig. 1 | Interface environments and OER performances at different temperatures. a–f** Structures of interface environments and the average distribution of water molecules along the [110] direction on TiO$_2$(110) at different water-density (ρ) and temperature conditions by controlling the external pressures, corresponding to 1 g/ml at 298 K, 1 g/ml at 373 K, 1 g/ml at 500 K, 0.85 g/ml at 500 K, 0.72 g/ml at 500 K, 0.52 g/ml at 500 K, respectively. Gray: Ti; red: O; white: H. **g** The water density distribution profiles of the first two layers. **h** Calculated OER rates at different water/TiO$_2$(110) interfaces as a function of temperature. The red curve: the

activity changes due to the interface environment change from the liquid state to the liquid-vapor coexistence state; the black dotted line: the activity changes with the interface environment unchanged (liquid state, the same as 298 K (l)). The 500 K (coexist) corresponds to the liquid-vapor coexistence state at 500 K at ρ = 0.52 g/cm$^3$. The shadow area shows the contribution of interface phase-change to activity enhancement. The inset shows the natural logarithmic plots of the OER rate at different water/TiO$_2$(110) interfaces as a function of reciprocal temperature.

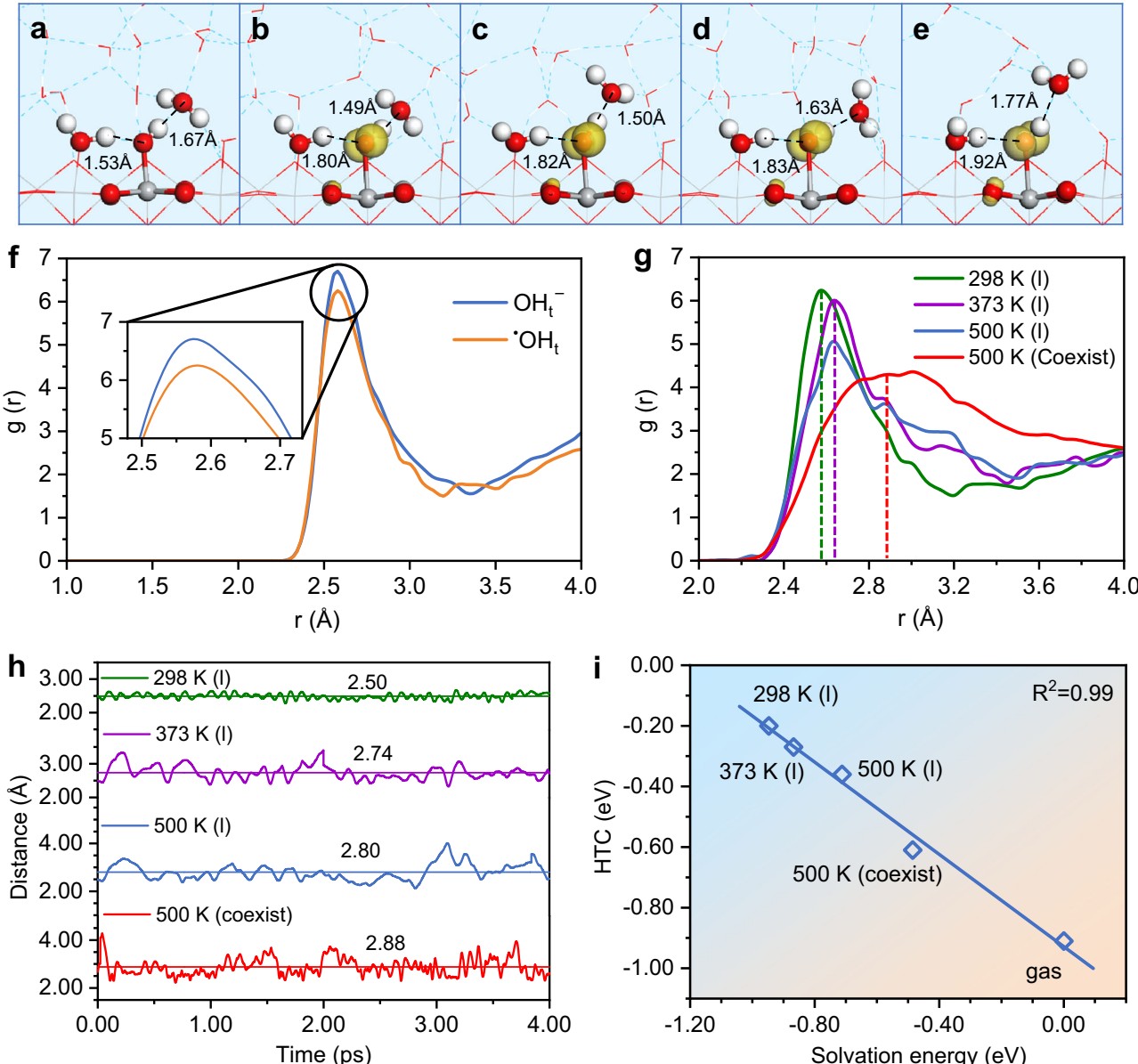

**Fig. 2 | The key characteristics of 'OH$_t$ at different interfaces.** Isosurfaces of the spin density (0.01 au) for the optimum interface structures: (**a**) hydroxide anion OH$_t^-$; (**b–e**) 'OH$_t$ radical at different conditions: 298 K (l), 373 K (l), 500 K (l), 500 K (coexist), respectively. **f** Radial distribution function (RDF), g(r)) for the OH$_t^-$ and 'OH$_t$ intermediates varying with the distance of O$_w$-O$_{OH}$ pair at 298 K (l). **g** g(r) for 'OH$_t$ radial at different conditions varying with the distance of O$_w$-O$_{OH}$ pair. **h** The distances of the O$_{OH}$ atom of 'OH$_t$ and the O$_w$ atom of nearest interface water as a function of simulation time. The horizontal line gives the average value of distance fluctuation. **i** The relationship between the hole trapping capacity (HTC) of 'OH$_t$ and the solvation energy of 'OH$_t$ at different interfaces.

## Characteristics of 'OH$_t$ radical and origin of OER activity modulated by interface environment

To unveil the mechanism of interface microenvironment effect on the OER rate, the degree of rate control (DRC)[27–29] of each elementary step was analyzed to identify the rate-limiting step. It was found that under ambient condition, the hole trapping at the terminal hydroxyl OH$_t^-$ to form 'OH$_t$ has the largest DRC value (Supplementary Fig. 5), indicating that this step is the rate-limiting step at 298 K (l). Comparing the DRC values of this step with those from other conditions, it can be seen that the formation of 'OH$_t$ is dominant in all cases. Therefore, further investigation was conducted to explore the key characteristics of the 'OH$_t$ radical.

Firstly, we analyzed the H-bond configurations of an adsorbed hydroxyl anion (OH$_t^-$) and a hydroxyl radical ('OH$_t$) at the water/TiO$_2$(110) interface. As shown in Fig. 2a, the adsorbed OH$_t^-$ forms a short H-bond (1.53 Å) as an acceptor and a long H-bond (1.67 Å) as a donor

from/to nearby water molecules. In contrast, the 'OH$_t$ radical can only form a long H-bond (1.80 Å) as an acceptor while maintaining a short H-bond (1.49 Å) as a donor with nearby water molecules. This observation demonstrates that OH$_t^-$ preferentially acts as an H-bond acceptor and is very hydrophilic, whereas the surface 'OH$_t$ predominantly act as an H-bond donor and becomes relatively hydrophobic[30]. Secondly, the interfacial surroundings of OH$_t^-$ and 'OH$_t$ at 298 K (l) were quantitatively compared using the radial distribution function (RDF, g(r); Fig. 2f). It was revealed that the first peak value of 'OH$_t$ at approximately 2.56 Å is lower than that of OH$_t^-$, indicating that 'OH$_t$ exhibits relatively hydrophobic characteristics compared to OH$_t^-$. This difference suggests that during the process of hole trapping at OH$_t^-$ to form 'OH$_t$ radical, the 'OH$_t$ intermediate would push the water molecules away, leading to a lower water density surrounding 'OH$_t$ relative to OH$_t^-$. This phenomenon can be attributed to a considerable reduction in the total charge of the central O atom in 'OH$_t$ radical (Supplementary Table 3).

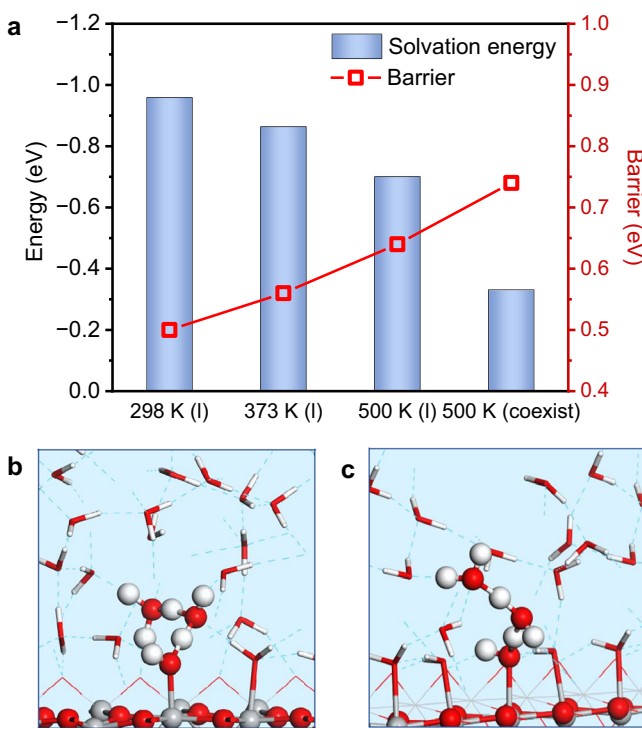

**Fig. 3 | Proton transfer in H₂O dissociation at different interfaces. a** The solvation energies of the surface adsorbed $H_2O_{ad}$ and the corresponding deprotonation barriers. **b, c** The transition state (TS) structure for proton transfer at 298 K (l) and 500 K (coexist), respectively.

With relatively hydrophobic nature of the ˙OH$_t$ radical, the influence of interfacial environments on the rate-limiting step (OH$_t^-$ + $h^+$ → ˙OH$_t$) can be disclosed. Firstly, according to Fig. 2b−e, the H-bonds surrounding ˙OH$_t$ gradually lengthen with increasing temperature. Additionally, the RDF of ˙OH$_t$ at different conditions (Fig. 2g) demonstrates that the peaks intensities decrease and shift towards higher distances as the temperature rises. Remarkably, at 500 K (coexist), the water density surrounding the ˙OH$_t$ radical decreases significantly and gives the largest distance between ˙OH$_t$ and the nearest interface water. Furthermore, when comparing the distance oscillations between the O atom of ˙OH$_t$ (O$_{OH}$) and the O atom of the nearest interfacial water (O$_w$) as a function of AIMD simulation time (Fig. 2h), it is evident that the oscillation amplitudes increase and the average O$_w$-O$_{OH}$ distance becomes longer as the temperature increases. Particularly, under the 500 K (coexist) condition, the water density decreases and the average H-O distance increases to 2.88 Å, which favors the formation of ˙OH$_t$ radical. Figure 2i shows the linear correlation between the hole trapping capacity (HTC) of ˙OH$_t$ and the solvation energy of ˙OH$_t$ radical (see details in Methods and Supplementary Table 4) at different interfaces, where HTC was calculated by the energy difference between the trapped hole and the self-trapped hole in the bulk of TiO$_2$[31]. It reveals that HTC becomes stronger as the solvation energy is reduced. This is the main reason that the OER rate increase more remarkably under the liquid-vapor coexisting condition.

It is worth noting that the ˙OH$_t$ radical formation is usually a slow step due to the low hole concentration under the ambient condition, thus hindering the whole OER progress[26]. In addition, the TiO$_2$ photocatalysts are hydrophilic upon illumination under standard conditions[32,33]. Therefore, improving the absolute value of HTC is effective to enhance the overall OER activity. This result could offer an additional rational to the fascinating experimental results of Chen et al. who discovered that the black hydrogenated TiO$_2$ holds the preeminent photocatalytic activities[34]: the black hydrogenated TiO$_2$

possesses the hydrophobicity[35,36] that might facilitate the ˙OH$_t$ radical formation to enhance photocatalytic activities in the system.

## Relationship between proton transfer and interface environment

There is no doubt that increasing the HTC blindly is not always effective. The highest HTC of ˙OH$_t$ (−0.91 eV) can be obtained if the gas phase condition is considered. However, it should be noted that the photocatalytic OER activity was negligible in the vapor phase environment under normal pressure, even though the temperature was increased[21]. To address this, the dissociation barriers of $H_2O_{ad}$ via proton transfer were analyzed. Figure 3a demonstrates that as the temperature increases, the solvation effect provided by the interfacial solution becomes weaker and the corresponding deprotonation barrier becomes higher. Figure 3b, c show the TS structure of the $H_2O_{ad}$ deprotonation process at 298 K (l) and 500 K (coexist), respectively. It can be observed that $H_2O_{ad}$ deprotonates in solution through a Grotthuss-type proton transfer mechanism and the detaching H$^+$ bounds to the nearby water at the interface (Supplementary Fig. 6). Importantly, we found that the deprotonation barriers of $H_2O_{ad}$ correlate well linearly with the solvation energy of $H_2O_{ad}$ (Supplementary Fig. 7). Therefore, once the density of water falls below a certain value, it becomes challenging for the proton of $H_2O_{ad}$ to touch the surrounding water, resulting in an increase in the deprotonation barrier. This observation helps explain why the dissociation of water is negligible under gas phase conditions, even with increased temperature[21]. On the other hand, when the water coverage and pressure are increased, the dissociation of water is enhanced[21,22]. Thus, the adequate density of the interface environment is the prerequisite to ensuring a successful reaction. The 500 K (coexist) interface microenvironment not only exhibits a sufficiently high HTC but also gives the appropriate water density, thereby leading to the superior OER activity.

## A strategy of manipulating the interface environment

Based on the unveiled intriguing interface microenvironment effect, here we propose a potential strategy to enhance OER performance without providing additional thermal energy. In this strategy, a partially hydrophobic substance (referred to as h-s in Fig. 4b), such as solid electrolyte, polyolefins, and other hydrophobic polymers, is proposed to be incorporated onto the catalyst surface to construct the appropriate interface water microenvironment. By introducing and tailoring the properties of hydrophobic substance and its interaction with the catalyst surface, the water density and distribution around the active sites can be possibly optimized to promote the OER performance.

In this work, hexafluoroacetone was chosen as a proof of concept to demonstrate the feasibility of the proposed strategy (see details in Supplementary Fig. 8). Figure 4b shows the optimized structure of the interfacial environment with hexafluoroacetone added. It can be observed that hexafluoroacetone creates some hydrophobic cavities for the reaction intermediates. Compared to the liquid-water/catalyst system (Fig. 4a), the presence of hexafluoroacetone leads to a thinner water density around the ˙OH$_t$ radical intermediate. Figure 4c illustrates the distance oscillations between O$_{OH}$ and the O$_w$ atom of nearest interfacial water as a -2.88 Å. This implies that this distance (-2.78 Å) does not impede proton transfer and can efficiently stabilize the species at the interface. Moreover, MPA-MD calculations demonstrate that the HTC of ˙OH$_t$ in the presence of hexafluoroacetone is reinforced to approximately −0.47 eV, compared to −0.20 eV at the pristine 298 K (l) condition. This indicates an enhanced hole trapping capacity. Finally, we calculated the elementary reaction steps of photocatalytic OER at the hexafluoroacetone modified water/TiO$_2$(110) interface and performed the microkinetic simulations to assess the overall activity using the obtained reaction energetics. As Fig. 4d

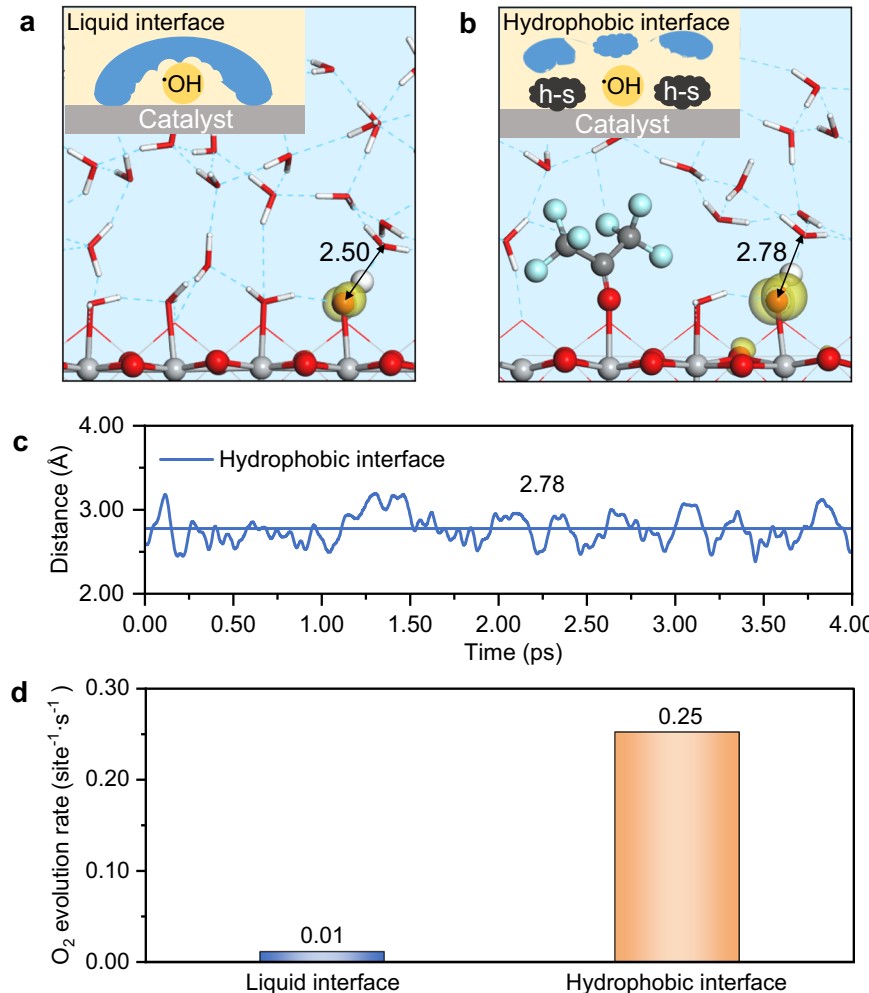

**Fig. 4 | Illustration of the strategy to improve OER activity via constructing hydrophobic interface.** Representative structures of the common water/catalyst interface (**a**) and the hydrophobic water/catalyst interface in the presence of hexafluoroacetone (**b**), where the changes of water distribution are illustrated as inserted. **c** The distances between the $O_{OH}$ atom of $\cdot OH_t$ and the $O_w$ atom of nearest interfacial water as a function of AIMD simulation time in the hydrophobic interface from the last ~4 ps. **d** Comparisons of the reaction activity of photocatalytic OER in the common and hydrophobic water/catalyst interfaces.

shows, the OER rate reaches ~0.25 site$^{-1}\cdot$s$^{-1}$, which gives a one-order-of-magnitude improvement compared to that (~0.01 site$^{-1}\cdot$s$^{-1}$) at the pristine water/TiO$_2$(110) interface at the same temperature (298 K).

## Discussion

It is worth stressing that tremendous progress has been made in understanding chemical reactions at the atomic level in the last hundred years, in particular in the last fifty years or so with the advances of modern experimental and first-principles simulation techniques, but the microenvironment is one of the unturned stones in the field. It is clear from our work that the microenvironment plays a huge role in affecting the chemical reactions: The reaction rate can be enhanced by several orders of magnitude by adjusting the microenvironment; the microenvironment change due to temperature change is even more important than the traditional temperature effect.

This work represents one of the first attempts to quantitively determine the temperature-dependent water/catalyst interface effect on the photocatalytic water splitting at the atomic level. We have found that the water/TiO$_2$(110) interface microenvironment dynamically affects the photocatalytic OER rate, and how the liquid-vapor coexistence phase achieves high OER performance has been unearthed. We revealed that the surface $\cdot$OH radical as the pivotal intermediate of photocatalytic oxygen evolution exhibits a unique relatively hydrophobic feature, resulting in the fact that additional

energy would be consumed to push the water network away in the formation of $\cdot$OH radical. The liquid-vapor coexistence environment can not only help the surface $\cdot$OH radical formation but also guarantee the functionality of water hydrogen-bond network for proton transfer, thus achieving the superior OER performance. Based on our findings, a simple and novel strategy has been demonstrated to enhance the photocatalytic OER activity by a factor of 25 under ambient condition by introducing hydrophobic hexafluoroacetone onto the TiO$_2$(110) substrate, which is one of the highest improvements in the literature. This study enhances our understanding of atomic-level photocatalytic reactions at liquid/solid interfaces and open avenues for designing catalytic systems that leverage interface microenvironments to achieve high catalytic performances.

## Methods

### Density functional theory (DFT) calculations

All spin-polarized calculations were performed in the Vienna ab initio simulation package (VASP)[37,38]. The DFT functional was utilized at the Perdew-Burke-Ernzerhof (PBE) level within the generalized gradient approximation (GGA), and the project augmented wave (PAW) method was used to represent the core-valence electron interaction[39]. The valence electronic states were expanded in plane-wave basis sets with an energy cutoff of 450 eV. The rutile TiO$_2$(110) surface was modeled by a four-layer $p(1 \times 4)$ periodical slabs with vacuum layer no less than

~15 Å. The reasons to choose such a model are given in Supplementary Fig. 9. Corresponding $1 \times 2 \times 1$ $k$-points mesh was used during optimizations. The Broyden-Fletcher-Goldfarb-Shanno (BFGS) quasi-Newton method[40] was applied to geometry relaxation until the Hellman-Feynman force on each ion was less than 0.05 eV/Å. The constrained optimization technique was applied to search the transition states (TS)[41,42], and the distance of the atoms that will form new bond is constrained at an estimated value. The TSs can be located via changing the fixed distance, and was verified when (i) all forces on the atoms vanish and (ii) the total energy is a maximum along the reaction coordinate but a minimum with respect to the rest of the degrees of freedom. To describe the van der Waals interaction in the system, the empirical DFT-D3 method with Becke-Jonson damping was used[43,44].

Considering the self-interaction error in $TiO_2$ system, the DFT + U method with the on-site Hubbard-type correction[31,45] was applied throughout this work. $U_{eff}$ values of 6.3 eV and 4.2 eV were adopted for O $2p$ and Ti $3d$ orbitals, respectively, which has been demonstrated in our previous studies with similar structures and reasonable energies to that in HSE06 functional within a reasonable timescale[26].

### Models for water/$TiO_2$(110) systems at different temperatures

To model the different water environments at water/$TiO_2$(110) interface, the ab initio molecular dynamics (AIMD) simulations were performed, in which lattice-matched pure bulk ices (containing 26 $H_2O$ molecules) were applied above the surface as an initial aqueous network (see geometry in Supplementary Fig. 10a). $k$-sampling was restricted to the $\Gamma$ point. A 0.5 fs movement was set for each step in the canonical (NVT) ensemble[46,47] with Nosé-Hoover thermostats. Over 9 ps MD simulations were performed, and all the simulations reach the equilibrium plateau after ~5 ps (see energy profile in Supplementary Fig. 10a).

Notably, to model the distinct microenvironment at water/$TiO_2$(110) interface under different conditions ($T$ = 298 K, 373 K, and 500 K), the volume was chosen to accommodate different densities of water by adjusting the height of Ar layer relative to $TiO_2$(110) surface (Supplementary Fig. 10b). The detailed parameters were presented in Supplementary Table 5, in which $h$ is the distance between Ar layer and surface $Ti_{5c}$ site deducting ~2 Å from the additional repulsion between Ar and water.

### Multi-point averaging molecular dynamics (MPA-MD) method

To calculate the energies of reactions occurring at the water/$TiO_2$(110) interface, the MPA-MD method was adopted, which was established in our previous study and has been demonstrated to calculate the solvation energy accurately in comparison to the energy obtained via the "slow-growth" method within the constrained AIMD framework[26] (see details in Supplementary Fig. 11). Specifically, we firstly selected ~10 samples every 0.2 ps from the equilibrium structure, and further optimized them to obtain the total energy of each structure ($E_{tot}$) using the DFT + U method. Next, considering the effect of different water network on $E_{tot}$, we deduct the total energy of water structures ($E_{water}$) and obtained the solvation-included energy of each intermediate ($E_{sol-included}$). Namely, the solvation-included energy of each intermediate $i$ can be expressed as $E_{sol-included}^i = E_{tot}^i - E_{water}^i$. Finally, we averaged the obtained solvation-included energies of all the samples in each intermediate. More details can be seen in our previous study[26]. In particular, it is worth noting that in order to obtain the samples of TS, the AIMD simulation was performed on a TS structure obtained using the constrained TS optimization technique. First, in this MD simulation, the reaction center of the TS structure was fixed, while all the other atoms (water solution and $TiO_2$ slab) were allowed to relax, with the aim of obtaining the equilibrated water structure that well accommodates the reaction center; then, each aqueous TS sample selected from the stabilized MD simulations was further re-optimized/refined for the energy statistical calculation.

### Calculations of interface energy

The interface energy ($E_{int}$(x)) of the adsorbate (x) is expressed by Eq. (1), which includes the contributions of adsorption energy ($E_{ads}$(x)) and the solvation energy of x ($E_{sol}$(x)), as shown in Eqs. (2) and (3), respectively.

$$E_{int}(x) = E_{tot} - E_{water + surf} - E_x \qquad (1)$$

$$E_{ads}(x) = E_{x/surf} - E_{surf} - E_x \qquad (2)$$

$$E_{sol}(x) = E_{int}(x) - E_{ads}(x) \qquad (3)$$

Here, $E_{tot}$ represents the total energy of the entire water and catalyst system with the adsorbate x adsorbed, $E_{water+surf}$ is the energy of the whole solution water and catalyst system excluding adsorbate x; $E_{x/surf}$, $E_{surf}$ and $E_x$ denote the single-point energies of adsorbate x adsorbed on the surface without the solution, the surface itself, and the adsorbate x, respectively. Supplementary Fig. 12 provides a further illustration of these energy components. To account for the configurational effect of interfacial water, these energies were obtained by averaging 10 samples selected from AIMD simulations. For specific energy information on the two key adsorbates ($H_2O_{ad}$ and $\cdot OH_t$) discussed in the main text, please refer to Supplementary Table 4.

### Calculations of free energy

The free energy of elementary reaction can be calculated with $\Delta G = \Delta H - T\Delta S + \Delta E_{ZPE}$, where $\Delta H$ is the enthalpy change; $T\Delta S$ is entropy change and can be obtained from the Handbook of Chemistry and Physics[48]. $\Delta E_{ZPE}$ is the zero-point-energies, which can be obtained through vibrational frequency calculations. For the surface reactions with no adsorption/desorption processes, the values of $T\Delta S$ and $\Delta E_{ZPE}$ are typically small, and thereby can be neglected[26]. The free energy changes of surface reactions can be approximately estimated from the enthalpy change. However, for the $H_2O$ adsorption (step 1) and $O_2$ desorption processes (step 7), entropy and zero-point-energies corrections should be considered and have displayed in Supplementary Table 6.

### Simulation of radical species

To simulate a photogenerated hole in the consideration of simplifying the complicated photoexcited system, we introduced an OH group as an electron acceptor on the bottom surface of the slab instead of extracting electrons, resulting in a hole in the system. The localization of hole is confirmed by the electronic structure analysis, including the site-projected magnetic moment and Bader charge analysis. It is worth noting that, in these strategies, the uncertain and relatively small electron-hole interaction that varies with the trapping center in the $TiO_2$ system was neglected, while ensuring the electroneutrality of the periodic system (eliminating the energy correction resulting from the presence of background counter charge). Such an approache had been used in our previous work[26,31,45,49].

### Hole migration barrier in the microkinetic modeling

To quantitatively determine the OER performance, the CATKINAS package[27] was used, which is a microkinetic simulation package developed by our group and widely used[50-53]. The hole concentration is $C_{h+} = 10^{-9}$ ML[54] and the hole migration barrier in the microkinetic modeling is estimated as $E_a = 0.30$ eV at 298 K (l)[55]. Generally, in the Marcus eletron transfer theory, the reduced reaction energy of surface species can contribute to lowering the electron transfer barrier. Additionally, Nurlaela and coworker reported that when the temperature is elevated, the mobility of holes and electrons is increased at 275−348 K, but the change in carrier density is negligible[56]. At higher temperatures (>400 K), the carrier concentration increases with temperature, which implies the hole transfer barrier reduces with

temperature. Moreover, the charge transfer to the surface OH fitting the Brønsted–Evans–Polanyi (BEP) relationship has been demonstrated, and the slope is around 0.5[19]. In this work, we estimated that the BEP slope of hole migration is 0.3. According to the HTC value, we can estimate the hole migration barrier at different interfaces, and the results are shown in Supplementary Fig. 13. It is worth noting that, according to our analysis, the concrete value of BEP slope only influences the absolute value of $O_2$ evolution rate under actual conditions, but the corresponding trend of $O_2$ evolution rate is unanimous with the trend shown Fig. 1h. Although the BEP slope changed, the superior OER performance is still achieved at 500 K (coexist) condition.

## Data availability
The data that support the findings of this study are available within the manuscript/ Supplementary Information and are also provided in Source Data file. Source data are provided with this paper.

## Code availability
The authors declare no any code access restrictions.

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

## Acknowledgements

This project was supported by National Key R&D Program of China (2021YFA1500700), NSFC (22202069, 92045303, 21703067, 21873028), Special Support by the China Postdoctoral Science Foundation (in front of the website) (2022TQ0106), the China Postdoctoral Science Foundation Funded Project (2022M721141), and the Fundamental Research Funds for the Central Universities.

## Author contributions

H.F.W. supervised the work; G.R. and M.Z. conducted the research, analyzed the data, and wrote the paper. P.H. and J.C. discussed the results and commented on the manuscript. G. R. and M. Z. contributed equally to this work.

## Competing interests

The authors declare no competing interests.
