## [Peer Review File · Nature Communications]

Bubble-water/catalyst triphase interface microenvironment accelerates photocatalytic OER via optimizing semi-hydrophobic OH radicalREVIEWER COMMENTS

Reviewer #1 (Remarks to the Author):

In this study, the authors utilized various theoretical methods, such as Density Functional Theory (DFT), Molecular Dynamics (MD), and microkinetic modeling, to investigate the atomic-level processes of photocatalytic OER on TiO₂(110). They discovered that changes in the interfacial microenvironment induced by temperature variations have a significant impact on the photocatalytic OER activity. Specifically, elevated temperatures or the presence of hydrophobic substances can disrupt the hydrogen-bond network, thereby enhancing the OER activity.

Since the reliability of computational results heavily relies on the computational methods employed, I agree to accept the manuscript for publication after considering the following concerns and provide a more comprehensive description of their computational techniques.

1. In line 372-376, the authors mentioned the “slow-growth” method within the constrained AIMD framework (Supplementary Fig. 7) when introducing the MPA-MD method. It would be beneficial to clarify the exact nature of the constrained MD and “slow-growth” approaches. Are these methods considered reliable? Additionally, it would be helpful to provide more detailed descriptions or references on these methods. Moreover, it would be interesting to compare the energy profiles of the H₂O deprotonation process obtained using the MPA-MD method with those shown in Supplementary Fig. 7.

2. The authors primarily employed the MPA-MD method, which requires MD calculations on the transition state structure. How did the authors determine and maintain the transition state structure during the MD process?

3. The TiO₂(110) surface exhibits layer dependent structural and energetic oscillations, which can cause significant deviations from experimental results when the number of TiO₂ layers is small [PANS, 2023, 120 (2) e2212250120]. How did the authors address or mitigate the impact of this issue in their study?

4. Could the authors provide a detailed explanation as to why the entropy change for H₂O adsorption (step 1) and O₂ desorption steps (step 7) is positive?

5. In line 260-261, it is mentioned that water dissociation under high-temperature vapor conditions becomes difficult. Does this difficulty also depend on the scarcity of adsorbed water molecules on the catalyst surface?

6. In Supplementary Figure 6b, is the grey sphere included in the “water thickness” indicative of Ar atoms? It would be helpful to clearly state this in the figure legend.

7. In Supplementary Table 2, the header indicates that the calculated values are free energies (ΔG), but the table employs enthalpy change (ΔH). Are the values presented in the table representative of free energy changes or enthalpy changes?

Reviewer #3 (Remarks to the Author):

In the article "Bubble-water/catalyst triphase interface microenvironment accelerates photocatalytic OER via optimizing semi-hydrophobic OH radical" Authors present a detailed study on the kinetics of the oxygen evolution reaction on titania and the key role of the hydrogen bonding network and of the OH radicals by using molecular dynamics simulations. The Authors further propose a possible methodology to generate a favorable microenvironment to improve the efficiency of the water oxidation reaction by adding hydrophobic molecules. The manuscript is well written and exhaustive, however a few comments, listed below, should be answered.

1) In the introduction at line 43 among the various phenomena affecting the reaction efficiency, the effect of micro and macro bubbles should be mentioned. Looking at the macro scale, a vapor is orders of magnitude less dense than a liquid, so the number of molecules close to the active sites and their probability to react is necessarily lower.

2) At line 112, Authors state "At the 500K (I) state stabilized at critical pressure". How the Authors were sure that the system was at critical pressure and how this pressure, which for water, is around 217.7 atm, was calculated?

3) I have some doubts about talking of "bubbles" in such small systems. In line 119, the Authors reported "the vapor bubbles diffuse stochastically...", it would be interesting a deeper analysis. Considering that the simulation box contains 24 water molecules, is it still possible to talk about bubbles? Rather, it seems to be a more disoriented hydrogen bonding network. To this hand, a more accurate analysis of the hydrogen bonds could be performed, for example comparing the average number of hydrogen bonds per water molecule in the 6 cases described (Figure 1 a-f).

4) It would be of interest to have a more detailed description of how the holes concentration (lines 158-159) and, thus, the oxygen evolution reaction, are related to the incoming light irradiation (spectrum and power of the light source/photon flux). This could be a helpful addition to Figure 1h and Figure 4d.

5) The manipulation of the microenvironment to improve the reaction kinetics is fascinating. What is the reason for using hexafluoroacetone? It would be interesting to know whether it could undergo decomposition due to light irradiation or due to the radicals generated during the water oxidation.

Responses to Reviewer's Comments

Responses to Reviewer #1

Comments: “In this study, the authors utilized various theoretical methods, such as Density Functional Theory (DFT), Molecular Dynamics (MD), and microkinetic modeling, to investigate the atomic-level processes of photocatalytic OER on TiO₂(110). They discovered that changes in the interfacial microenvironment induced by temperature variations have a significant impact on the photocatalytic OER activity. Specifically, elevated temperatures or the presence of hydrophobic substances can disrupt the hydrogen-bond network, thereby enhancing the OER activity. Since the reliability of computational results heavily relies on the computational methods employed, I agree to accept the manuscript for publication after considering the following concerns and provide a more comprehensive description of their computational techniques.”

Response: We really appreciate the referee's positive comments and valuable suggestions. The detailed responses to specific comments are given below.

Q1. “In line 372-376, the authors mentioned the “slow-growth” method within the constrained AIMD framework (Supplementary Fig. 7) when introducing the MPA-MD method. It would be beneficial to clarify the exact nature of the constrained MD and “slow-growth” approaches. Are these methods considered reliable? Additionally, it would be helpful to provide more detailed descriptions or references on these methods. Moreover, it would be interesting to compare the energy profiles of the H₂O deprotonation process obtained using the MPA-MD method with those shown in Supplementary Fig. 7.”

Response: We thank the referee for this valuable concern and suggestion. To take the point of the referee, the corresponding discussions have been added in the revised Supplementary Information (Supplementary Fig. 11; Pages S16-S18).

Firstly, the constrained MD method was established on the basis of thermodynamic integration of the free-energy gradient within the statistical mechanics framework (*J. Chem. Phys.* **1998**, 109, 7737; *J. Phys. Chem. B* **2000**, 104, 823; *J. Phys. Condens. Matter* **2008**, 20, 064211; *J. Phys. Chem. C* **2021**, 125, 10974). It is a well-accepted approach to accurately calculate the free energy change (including reaction barriers). In the constrained MD method, the Nosé-Hoover thermostat and NVT ensemble were usually used. The free energy difference (ΔF) between state (1) and (2) is obtained based on the identification of the gradient $(\partial F/\partial \xi)_{\xi^*}$ determined at each constrained distance $\xi(1 \rightarrow 2)$, where the $(\partial F/\partial \xi)_{\xi^*}$ stands for the statistical average of $(\partial F/\partial \xi)$, and ξ is the reaction coordinate typically corresponding to a key geometric parameter linking state (1) and (2). As shown in eqn-R1, ΔF can be computed by integrating free energy gradients over ξ along the reaction path (1 \rightarrow 2):

$$\Delta F_{1 \rightarrow 2} = \int_{\xi(1)}^{\xi(2)} \left(\frac{\partial F}{\partial \xi} \right)_{\xi^*} d\xi \quad (\text{eqn-R1})$$

In our constrained MD simulation for calculating the barrier of H₂O deprotonation process, the reaction coordinate ξ is equal to the difference between the HO_{ad}-H_{ad} bond distance (l_1) in H₂O_{ad} and the H_{ad}-O_n distance l_2 (O_n is the oxygen atom of the nearest interface water molecule), that is $\xi=l_1-l_2$ as illustrated in Fig. R1a. As shown in Fig. R1b, the ξ value is stretched gradually from -0.65 to 1.0 Å during the H₂O deprotonation process. For each fixed ξ , we performed long-time MD simulation in NVT ensemble (T =298 K) until quasi-equilibrium state was achieved. All the interatomic forces along the reaction coordinate, which corresponds to the free energy gradients, can be obtained statistically. Then the free energy change can be obtained by integrating these free energy gradients.

Overall, the constrained MD is an accurate but very time-consuming method to calculate the complex OER network at the water/TiO₂(110) interface. Alternatively, we developed the MPA-MD method to deal with the aqueous systems, and the reaction energetics were thoroughly tested in our previous work, including the reaction barriers, to verify the reliability of MPA-MD method (*Nat. Catal.* **2018**, 1, 291). As shown in Table R1, five kinds of aqueous interface reactions were compared between MPA-MD and the state-of-the-art constrained MD method. The MPA-MD method gives very similar results to the constrained MD method in all cases. In particular, for the H₂O deprotonation reaction, both methods give the comparable barriers (0.50 versus 0.55 eV; see Fig. R1c), demonstrating the feasibility of the MPA-MD method in studying the OER mechanism at the water/TiO₂(110) interface.

Fig. R1 **a** The definition of the reaction coordinate, equal to the difference between H_{ad}-O_{ad} bond distance l_1 in H₂O_{ad} and the H_{ad}-O_n distance l_2 (O_n is the oxygen atom of nearest water in liquid). **b-d** Energy profiles of H₂O deprotonation process at H₂O/TiO₂(110) interface using the constrained MD method (**b**), the MPA-MD method (**c**), and the slow-growth method (**d**).

Table R1. Comparisons of reaction barriers (E_a) using the MPA-MD and constrained MD methods.

Reaction in solutions	E_a (eV) (MPA-MD)	E_a (eV) (constrained MD)
$\text{Pt}(\text{NH}_3)_2\text{Cl}_2 + \text{H}_2\text{O} \rightarrow \text{Pt}(\text{NH}_3)_2\text{Cl}(\text{H}_2\text{O})^+ + \text{Cl}^-$	0.78	0.75 (exp. 0.84-1.07) ^a
$\text{Na}_n\text{Cl}_n \rightarrow \text{Na}_n\text{Cl}_{n-1}^+ + \text{Cl}^-$	0.15	0.11 ^b
$\text{O}_2 \rightarrow 2\text{O}^*$ on Pt (111)	0.43	0.39
$\text{O}_2 + \text{H}^* \rightarrow \text{OOH}^*$ on Pt (111)	0.60	0.58
$\text{H}_2\text{O}^* \rightarrow \text{OH}^- + \text{H}^+$ on TiO_2 rutile(110)	0.50	0.55

Ref.^a (*J. Chem. Phys.* **2006**, 125, 091101), ref.^b (*Phys. Chem. Chem. Phys.* **2011**, 13, 13162)

Secondly, the slow-growth based AIMD method is another effective way to calculate the free energy profile, which has been successfully used to capture the varying H-bonding networks during the aqueous reaction (*J. Phys. Chem. B* **1997**, 101, 7877; *Phys. Rev. Lett.* **1997**, 78, 2690; *J. Am. Chem. Soc.* **2020**, 142, 5773). Compared to the common constrained MD method, the reaction coordinate (ξ) in the slow-growth method is changed linearly from state (1) to state (2) with a constant and very small transformation velocity $\dot{\xi}$. This is the biggest difference from the common constrained MD approach. The resulting free energy difference needed to perform a transformation from state (1) to state (2) can be computed as:

$$\Delta F_{1 \rightarrow 2} = \int_{\xi(1)}^{\xi(2)} \left(\frac{\partial F}{\partial \xi} \right)_{\xi^*} d\xi = \lim_{\dot{\xi} \rightarrow 0} \int_{\xi(1)}^{\xi(2)} \left(\frac{\partial F}{\partial \xi} \right) \cdot \dot{\xi} dt \quad (\text{eqn-R2})$$

where F is the free energy calculated at the coordinate ξ which evolves with t , and $\partial F/\partial \xi$ is calculated along the MD trajectory by the SHAKE algorithm (*J. Comput. Phys.* **1997**, 23, 327). Overall, the slow-growth method has a relatively lower computational cost at a similar (or slightly worse) accuracy than the common constrained MD method, which was thus also used to quickly determine the reaction coordinate prior to the common constrained MD method.

Specifically, for the H_2O deprotonation process, $\xi = l_1 - l_2$ was similarly chosen as the collective variable (CV) as shown in Fig. R1d, and a very small value $\partial \xi$ of 0.0005 Å was used in practice for each MD step. It is worth mentioning that the shorter step size for describing the slow-growth process along the reaction coordinate was also tested, which verified the validity of the $\partial \xi$ value. Notably, the slow-growth approach is available in the VASP code, and we used the standard exponentially weighted moving average (EWMA) method to process the average value of $\partial F/\partial \xi$. An example of the raw output data from VASP and the integrated free energy profile is demonstrated in Fig. R1d. The barrier of H_2O deprotonation process with this approach is 0.52 eV, which is close to the value obtained from MPA-MD method (0.50 eV).

Q2. “The authors primarily employed the MPA-MD method, which requires MD calculations on the transition state structure. How did the authors determine and maintain the transition state structure during the MD process?”

Response: We thank the referee for this valuable comment. In the MPA-MD method, the starting structure for each transition state (TS) was searched using the *constrained TS optimization technique* (see Method). Taking the transition state of water dissociation (TS1 in Supplementary Fig. 6) as an example, we first obtained an optimized liquid/solid interface structure with a water molecule adsorbed on the surface, namely the initial state for water dissociation (IMS-1 in Supplementary Fig. 6). The TS structure was then searched by gradually stretching the O-H bond while monitoring the force on the O/H atoms until the threshold of 0.05 eV/Å was reached, *i.e.*, obeying the *constrained TS optimization technique*. Next, MD simulations were performed on the obtained TS structure with the reaction centre of the TS structure fixed, while all the other atoms (water solution and TiO₂ slab) were allowed to relax during the MD simulations. This strategy is applied for the purpose of obtaining the equilibrated water structure that well accommodates the reaction centre with reasonably accurate solvation energies. Finally, we selected a series of aqueous configurations every 0.2 ps from the stabilized MD simulations and further re-optimized/refined the TS structure in each aqueous configuration.

To address the referee's question, we have made some changes to make it more clear in the Method section in the revised manuscript (line 354 on Pages 16-17).

Q3. “The TiO₂(110) surface exhibits layer dependent structural and energetic oscillations, which can cause significant deviations from experimental results when the number of TiO₂ layers is small [PNAS, 2023, 120 (2) e2212250120]. How did the authors address or mitigate the impact of this issue in their study?”

Response: We thank the referee for this valuable comment. Indeed, the TiO₂(110) surface exhibits layer-dependent structural and energetic oscillations, predominantly characterized by odd-even oscillation with respect to the number of layers in a slab. This observation is largely determined by the atomic structure of TiO₂(110) and is a long-standing concern in the literature. To mitigate the impact of this issue, and also to ensure that the project progresses within a reasonable timescale, there are three reasons for us to choose the 4-layer $p(4\times 1)$ model:

Firstly, the use of 4-layer $p(4\times 1)$ model yields reasonably accurate results within a manageable computational time frame. As illustrated in Fig. R2a, we compared the results for the water dissociation barrier (E_a) and enthalpy change (ΔH) on the rutile TiO₂(110) surface against the number of TiO₂ layers. We can see that the 4-layer model gives similar results to those from the 10-layer slab. Also, the results from the 4-layer model are close to those from the 6-layer and 8-layer models. It is important to note that although the use of thicker slabs unquestionably gives more reliable results, the computational workload increases exponentially with thickness. Therefore, within a reasonable margin of error, the 4-layer model is the more economically viable choice.

Secondly, the 4-layer $p(4\times 1)$ model has been consistently used in the published theoretical and computational work, as supported by a number of studies in the field of photocatalysis (*J. Am. Chem. Soc.* **2015**, 137, 9146; *Nat. Catal.* **2018**, 1, 291; *ACS Catal.* **2017**, 7, 2374; *JACS Au*

2022, 2, 188). Specifically, we compared the energy profile of water dissociation on the 4-layer $p(4\times 1)$ surface with the experimental data reported by Wang et al. (*PNAS* 2017, 114, 1801), demonstrating the agreement between the calculated results ($E_a=0.37$ eV and $\Delta H=0.16$ eV) and experimental data ($E_a=0.36$ eV and $\Delta H=0.035$ eV). This alignment underscores the reliability of our chosen model in capturing the energetics of this process.

Thirdly, one of the primary focuses in this current work is on accurately simulating the aqueous states at different conditions and obtaining the reliable reaction energies. According to the reference recommended by the referee (*PNAS* 2023, 120, e2212250120), the water density distributions are very similar for the different slabs from 4 to 16 layers and overlap each other almost completely (Fig. R2b). This result further confirms that the 4-layer structure can be used to simulate a realistic solid-liquid interface system.

To take the referee's suggestion, the corresponding discussions have been added in the revised manuscript (line 314 on Page 15) and revised Supplementary Information (Supplementary Fig. 9; Pages S13-S14).

Fig. R2 a Comparison of energy barrier (E_a) and enthalpy change (ΔH) for water dissociation on rutile $\text{TiO}_2(110)$ versus the number of TiO_2 layers. The horizontal dotted line gives the value of 10-layer slab. Inset structures are the initial state (IS), transition state (TS), and final state (FS), respectively. Grey, red, and white balls represent Ti, O, and H atoms, respectively. **b** Water density distributions as a function of the number of O-Ti-O tri-layers on the rutile $\text{TiO}_2(110)$ slab (taken from *PNAS* 2023, 120(2), e2212250120).

Q4. "Could the authors provide a detailed explanation as to why the entropy change for H_2O adsorption (step 1) and O_2 desorption steps (step 7) is positive?"

Response: We thank the referee for pointing out this careless input error.

For the H_2O adsorption step (step 1): $\text{H}_2\text{O}(\text{sol}) + * \rightarrow * \text{H}_2\text{O}$, the entropy change is determined from $\Delta S_1 = S_{\text{FS}} - S_{\text{IS}} = S(* \text{H}_2\text{O}) - S(\text{H}_2\text{O}(\text{sol}))$, the value of which is thus negative. For the O_2 desorption step (step 7): $* \text{O}_2^- + h^+ \rightarrow \text{O}_2(\text{aq}) + *$, the entropy change is calculated from $\Delta S_7 = S_{\text{FS}} - S_{\text{IS}} = S(\text{O}_2(\text{aq})) - S(* \text{O}_2^-)$, the value of which is positive.

This mistake has been corrected in Supplementary Table 6 in the revised Supplementary Information (Page S26).

Q5. “In line 260-261, it is mentioned that water dissociation under high-temperature vapor conditions becomes difficult. Does this difficulty also depend on the scarcity of adsorbed water molecules on the catalyst surface?”

Response: We thank the referee for this valuable comment/question. We agree with the referee that the coverage of water on the surface does indeed play a role in water dissociation, which can modulate the interactions between the catalyst surface and the adsorbed water molecules (usually referred to as the coverage effect). In general, decreasing the water coverage could make the water dissociation more difficult, as also mentioned in our original main text (line 243 on page 12) that “On the other hand, when the water coverage and pressure are increased, the dissociation of water is enhanced (*Catal. Lett.* **2008**, 125, 376; *Angew. Chem. Int. Ed.* **2019**, 58, 17751).” We would like to point out the following: (i) Under the high-temperature conditions, the coverage effect of water molecules has been incorporated, as the adsorption/desorption of water is allowed in the high-temperature MD simulation to model the dynamic interfacial environment. (ii) In this work, we demonstrate that the primary influencing factor is the variation of the aqueous environment, with the coverage of water molecules being a secondary factor. The specific reasons are as follows:

Firstly, it is worth noting that during the high-temperature MD simulations, the surface adsorbed water molecules (*except for the reactant waters*) were allowed to evolve together with the interface water solution, which allows the possible water desorption in the free-energy landscape, and thus the contribution of water coverage has been included in our energy calculations. In this work, the dissociation barriers of $\text{H}_2\text{O}_{\text{ad}}$ via proton transfer at different interfaces have been analysed. Fig. 3a in the main text shows that when the solvation effect provided by the interfacial solution becomes weaker, the corresponding deprotonation barrier becomes higher. It can be observed that the barrier changes from 0.50 eV to 0.74 eV for 298 K(l) to 500 K(coexist) systems, with the solvation energy changing from -0.96 eV to -0.33 eV correspondingly. More importantly, the deprotonation barriers correlate well linearly with the solvation energies of $\text{H}_2\text{O}_{\text{ad}}$ (Supplementary Fig. 7). Considering that the density of water is 0.52g/cm^3 at 500 K(coexist), the energy barrier is increased by 0.24 eV compared to the state at 298 K(l), while under the high-temperature vapor conditions, the water is orders of magnitude less dense than the condition at 500 K(coexist). Therefore, it becomes more difficult for the deprotonation process of $\text{H}_2\text{O}_{\text{ad}}$ under the high-temperature vapor conditions, resulting in a higher barrier and even impeding the proton transfer.

Secondly, following the referee’s advice, we tested the coverage effect on water dissociation over $\text{TiO}_2(110)$ (with the solvation contribution of aqueous environment decoupled), and we found that the coverage effect of $\text{H}_2\text{O}_{\text{ad}}$ is relatively small, as shown in Fig. R3. Specifically, since the MD simulations show that there is typically a 75% (3/4 monolayer) coverage of water adsorbed on $\text{TiO}_2(110)$ under the aqueous condition (Supplementary Fig. 2), we examined the water coverages below 75% as a possible response to the high temperature condition. It can be seen from Fig. R3 that as the water coverage decreases from 75% to 25%, the overall trend of

the water dissociation barriers increases, with a slight oscillatory pattern; it first increases significantly from 75% to 50% and then decreases slightly from 50% to 25%. This result is influenced by the hydrogen bonding effects between the adjacent H₂O molecules and intricate long-range adsorbate-adsorbate repulsion (including the competing adsorption-induced substrate relaxation). For example, when comparing the transition-state structures B (50%) and C (75%), it is evident that the absence of H-bond *ii* (see Fig. R3) between O₁ and H₆, which stabilizes the O atom (in orange) of H₂O_{ad}, makes the dissociation of O₁-H₃ bond more difficult, thereby increasing the barrier of water dissociation. Furthermore, as can be seen from structure B (50%) to A (25%), the absence of H-bond *i* between O₅ and H₄ strengthens the O₁-H₄ bond and consequently weakens the O₁-H₃ bond. In addition, it might be worth noting that we also studied the case of 100% coverage (see structure D in Fig. R3), which exhibits a slightly higher water dissociation barrier compared to the 75% case, largely due to similar H-bond effect. Despite these intricate chemical interactions, the magnitude of the barrier variation is relatively small (less than 0.07 eV), demonstrating that the overall interaction changes relatively little as the coverage changes.

Fig. R3 The correlation between the water dissociation barrier and the water coverage on TiO₂(110) in the absence of liquid. Inserted structures A-D are the transition states at different water coverages. The grey and red balls stand for Ti_{5c} and O atoms, respectively, while the orange ball is the O atom of the dissociating H₂O; the white ball represents H atom, and the blue ball is the dissociated H to be bound with O_{br}.

Q6. “In Supplementary Fig. 6b, is the grey sphere included in the “water thickness” indicative of Ar atoms? It would be helpful to clearly state this in the figure legend.”

Response: We appreciate your valuable advice. The grey spheres included in the “water thickness” indeed indicate Ar atoms. To take the point of the referee, it has been added in the figure caption of Supplementary Fig. 10b in the revised Supplementary Information (Page S15).

Q7. “In Supplementary Table 2, the header indicates that the calculated values are free energies (ΔG), but the table employs enthalpy change (ΔH). Are the values presented in the table representative of free energy changes or enthalpy changes?”

Response: We thank the referee for his/her meticulous review of our manuscript. The calculated values are free energies (ΔG) in the **Supplementary Table 2**. The label mistake has been corrected in the revised Supplementary Information (**Page S22**).

Responses to Reviewer #3

Comments: *“In the article “Bubble-water/catalyst triphase interface microenvironment accelerates photocatalytic OER via optimizing semi-hydrophobic OH radical” Authors present a detailed study on the kinetics of the oxygen evolution reaction on titania and the key role of the hydrogen bonding network and of the OH radicals by using molecular dynamics simulations. The Authors further propose a possible methodology to generate a favorable microenvironment to improve the efficiency of the water oxidation reaction by adding hydrophobic molecules. The manuscript is well written and exhaustive, however a few comments, listed below, should be answered.”*

Response: We thank the referee’s positive comments and valuable suggestions. The detailed responses to the specific comments are given below.

Q1. “In the introduction at line 43 among the various phenomena affecting the reaction efficiency, the effect of micro and macro bubbles should be mentioned. Looking at the macro scale, a vapor is orders of magnitude less dense than a liquid, so the number of molecules close to the active sites and their probability to react is necessarily lower.”

Response: We appreciate the referee for this kind suggestion. The point of the referee has been taken. The corresponding discussions have been added to the Introduction section (line 42 on page 3) in the revised manuscript. Indeed, at the macro-scale, a vapor in the macro-bubble state has low density, which would evidently affect the probability of adsorption and reaction events. This is also one reason of the low activity of vapor/TiO₂(110) interface for OER at high temperature.

In general, as the temperature increases, the hydrogen bond network in the aqueous system would be distorted, reconstructed, and broken, gradually leading to the formation of nano-sized cavities (we called them as ‘micro-bubbles’) in the aqueous solution. With the continued increase in temperature, the macro-bubbles would gradually form and eventually transform into vapor. During this process, the water density at the solid-liquid interface gradually decreases. Especially in the vapor state, one can envisage that the density of water molecules is several orders of magnitude lower than that in the liquid condition, and the number of molecules close to the active sites and their probability to react are lower. In addition, in this study we also demonstrated computationally that the proton transfer channel would be obstructed and thus inhibits the whole OER. To the best of our knowledge, there was no atomic-level understanding of how the micro-bubble environment influences the OER process in the photoexcitation condition, which is also one of the main highlights of our current study.

Based on referee’s suggestion, the corresponding discussions have been added to the revised manuscript (line 42): “... possibly due to the fact that the vapor phase in a macro-/micro-bubble environment is significantly less dense than liquid, and the number of molecules close to the active sites and the reaction probabilities are lower.”

Q2. "At line 112, Authors state "At the 500K (l) state stabilized at critical pressure". How the Authors were sure that the system was at critical pressure and how this pressure, which for water, is around 217.7 atm, was calculated?"

Response: We appreciate the referee's valuable comment. It was our fault that the description of "At the 500 K(l) state stabilized at critical pressure" was inaccurately phrased, leading to the misunderstanding. When we referred to the "at 500 K(l) state stabilized at critical pressure", we were not discussing the actual pressure at *the critical point*, which is around 217.7 atm. Instead, we were considering an idealized model at 500 K without a phase transition for the liquid water. In this model, the water is kept in the liquid-phase state by exerting high pressure to prevent vaporization, ensuring a constant water density of 1g/cm^3 . Based on this ideal model, this condition represents the boiling point of 500 K under approximately 27.8 atm, which was calculated from boiling point calculator (<https://www.calctool.org/thermodynamics/boiling-point>).

Therefore, it has been changed to "At the 500 K (l) state stabilized at a high pressure (~27.8 atm)" in the revised manuscript (line 113 on page 6).

Q3. "I have some doubts about talking of "bubbles" in such small systems. In line 119, the Authors reported "the vapor bubbles diffuse stochastically...", it would be interesting a deeper analysis. Considering that the simulation box contains 24 water molecules, is it still possible to talk about bubbles? Rather, it seems to be a more disoriented hydrogen bonding network. To this hand, a more accurate analysis of the hydrogen bonds could be performed, for example comparing the average number of hydrogen bonds per water molecule in the 6 cases described (Figure 1 a-f)."

Response: We thank and agree with the referee for this valuable comment. The context of "the vapor bubbles diffuse stochastically..." is indeed a general (*or conceptual*) description of such a system. In the case of 500 K (coexist), the density of water molecules is already lower than that of the typical liquid water. Particularly under the condition depicted in Fig. 1f of the main text, the volume occupied by water molecules has nearly doubled, resulting in a significant 48% decrease in water density. In terms of the structural configuration, water molecules in typical liquid phase interact with adjacent molecules, forming six-, five- or four-member ring structures. However, it is evident that these original multi-membered ring structures are disrupted in the case of 500 K (coexist), resulting in *cavities* that are much larger than the multi-membered ring structures. Under such a condition, some water molecules are so far apart that they have lost their hydrogen bonding interactions. Hence, we used the term 'bubbles' to conceptually represent the cavities resulting from the broken hydrogen-bond network at 500 K (coexist). Overall, we acknowledge that the term 'bubbles' serves as a *vivid* definition to describe these cavities due to the use of a relatively limited-size model, as employing a larger slab and more water molecules to directly model the observable 'bubble' through AIMD simulation is computationally challenging. In response to the referee's suggestion, we have changed the "bubbles" into "micro-

bubbles"; moreover, we have added the hydrogen bond number/length analysis as requested in the revised manuscript (see below) (line 121 on Page 6) and Supplementary Information (Supplementary Fig. 1; Page S2).

Fig. R4 reveals the numbers and lengths of hydrogen bonds as a function of the distance from the interface in six cases from the statistical analysis. As the volume increases, the average number of H-bonds per water molecule gradually decreases. Particularly, at the first and second water layers of the liquid/catalyst interface, there is a significant reduction in the number of H-bonds at 500 K (coexist). This demonstrates that the hydrogen bond network is destroyed to some extent once the liquid-gas transition begins to occur. Furthermore, as the H-bonds are disrupted, the average length of the hydrogen bonds also increases (Fig. R4b).

Fig. R4 Comparison of the distributions of H-bonds per water molecule along the interface. **a** Distribution of the average number of H-bonds per water molecule. **b** Distribution of the average length of H-bonds per water molecule.

Q4. “It would be of interest to have a more detailed description of how the holes concentration (lines 158-159) and, thus, the oxygen evolution reaction, are related to the incoming light irradiation (spectrum and power of the light source/photon flux). This could be a helpful addition to Figure 1h and Figure 4d.”

Response: We thank the referee for this in-depth discussion and kind suggestion. We agree with the referee that the incoming light irradiation, including its spectrum, power, and photon flux, would affect the hole concentration, which in turn influences the rate of oxygen evolution reaction. Following the referee’s suggestion, the further analysis of the hole concentration is as follows:

In Fig. R5, we highlight that the concentration of holes reaching the surface has a significant impact on the OER rate. Specifically, we have investigated how the hole concentrations affect the OER rate under four typical conditions (298 K(l), 373 K(l), 500 K(l) and 500 K(coexist)) with the hole concentration ranging from 10^{-10} to 1 ML, as illustrated in Fig. R5. It can be seen the following: (i) The OER rates increase significantly from the very low concentration of the

surface-reaching hole and reach a plateau at a certain threshold of the hole concentration. The threshold values become gradually smaller from 298 K(l)→373 K(l)→500 K(l)→500 K(coexist), indicating the relatively small dependence of hole concentration on the liquid-vapor coexisting interface environments. (ii) Compared with the other three conditions, it consistently exhibits better OER activity under the 500 K (coexist) condition with varying hole concentration, aligning with the corresponding activity trend described in Fig. 1h of the main text.

Fig. R5 The logarithmic plots of the OER rates at different water/TiO₂(110) interfaces as a function of hole concentration.

Here we take the case of 500 K(coexist) as an example to further illustrate the effect of hole concentration. When the hole concentration is in the range from 10^{-10} to 10^{-4} ML, an increase in hole concentration leads to a higher rate of OER. When the hole concentration exceeds 10^{-4} ML, the OER rate reaches a plateau and the rate-determining step shifts, because at a higher hole concentration, the formation of $\cdot\text{OH}_t$ via trapping holes at the terminal OH^- becomes easier and is no longer the rate-determining step. Moreover, the presence of micro-bubbles under the liquid-vapor coexistence conditions slows down the water dissociation reaction and makes it earlier to be the rate-determining step (relative to the 298 K(l), 373 K(l) and 500 K(l) conditions). In this case, increasing the concentration of surface-reaching holes cannot further increase the reaction rate.

To take the points of the referee, the corresponding description has been added in Supplementary Information (Supplementary Fig. 4; Pages S6-S7), and in the revised manuscript (line 168 on Page 8) we have added some sentences for clarification: “It is worth noting that, we have performed an investigation of the influence of hole concentration on the OER rate, covering a range from 10^{-10} to 1 ML (Supplementary Fig. 4). Remarkably, the system consistently exhibits excellent OER activity under the 500 K (coexist) condition, which is in agreement with the activity trend described in Fig. 1h.”

Q5. “The manipulation of the microenvironment to improve the reaction kinetics is fascinating. What is the reason for using hexafluoroacetone? It would be interesting to know whether it could undergo decomposition due to light irradiation or due to the radicals generated during the water oxidation.”

Response: We thank the referee for the positive comment and for the valuable question. Indeed, the stability of the hydrophobic substance in the *in situ* photocatalytic condition is an important issue for the real application. It is worth mentioning that we use hexafluoroacetone as a proof of concept to demonstrate the effect of hydrophobic microenvironment on photocatalytic OER. More specifically, hexafluoroacetone was used for the following basic reasons: (i) It contains -CF₃ groups, which has strong hydrophobicity. (ii) It has a C=O group, which allows it to be adsorbed on the catalyst surface. Moreover, its adsorption energy is relatively lower than that of water molecule, and thus it will not excessively occupy the reaction sites. (iii) It is thermodynamically stable and does not chemically react with water molecules. For example, the enthalpy change of the reaction $\text{H}_2\text{O} + (\text{CF}_3)_2\text{C}=\text{O} \rightarrow (\text{CF}_3)_2\text{CHO}^\bullet + \bullet\text{OH}$ is strongly endothermic ($\gg 2$ eV). (iv) Regarding the photoexcitation effect, we evaluated the ability of C₃F₆O to trap holes, and it was found that C₃F₆O has a significantly weak hole trapping capability (HTC) of about -0.24 eV, which is much weaker than that of OH⁻ (-0.91 eV). As shown in the density of states (DOS) in Fig. R6a, the highest occupied state of O in C₃F₆O is much lower, as low as -1.30 eV, compared to the O of OH⁻ (located at -0.5 eV). Therefore, C₃F₆O itself is less likely to be photoexcited under photocatalytic condition, making it difficult to decompose.

To be honest, we previously overlooked the possibility of hexafluoroacetone conversion assisted by the surface OH[•] radicals under the reaction condition. Following the referee’s suggestion, we have investigated the oxidative conversion of hexafluoroacetone by the surface OH[•] radicals, as shown in Fig. R6c.

Specifically, as (CF₃)₂C=O cannot be activated via trapping hole (as shown in Fig. R6a), the coupling of (CF₃)₂C=O with OH[•] radical ((CF₃)₂C=O^{*} + ^{*}OH[•] → (CF₃)₂C(OH)O^{*}, i.e., process 1→3) as the prerequisite reaction to activate (CF₃)₂C=O was calculated, including the possible dehydrogenation conversion of (CF₃)₂C(OH)O^{*} to the carboxylate species ^{*}(CF₃)₂COO⁻. As shown in Fig. R6c, the whole process (1→4) is exothermic but gives an effective barrier as high as 1.01 eV; this indicates that hexafluoroacetone could be oxidized by OH[•] radical thermodynamically, but kinetically unfavourable at typical room temperature. In other words, hexafluoroacetone should be relatively stable, although the long-term stability could be an issue.

Based on referee’s suggestion, the corresponding discussions have been added to the revised Supplementary Information (Supplementary Fig. 8; Pages S11-S12), in which we have particularly emphasized the importance of considering the long-term stability when selecting the other hydrophobic substance for realistic application in further studies.

REVIEWER COMMENTS

Reviewer #1 (Remarks to the Author):

The authors have fully addressed the concerns from the reviewers. I have no more comments. It can be accepted as is.

Reviewer #3 (Remarks to the Author):

I appreciated the rebuttal from the Authors and all my questions have been answered, therefore I suggest to accept the manuscript.

Reviewer #4 (Remarks to the Author):

This work presents a comprehensive computational investigation of the atomic-level processes of photocatalytic oxygen evolution reaction (OER) on the aqueous TiO₂(110) interface, utilizing first-principles density functional theory calculation, molecular dynamics simulation, and microkinetic analysis. The authors observed that the crucial intermediate OH• radicals involved in the photocatalytic OER exhibit semi-hydrophobic properties. They present quantitative evidence indicating that the temperature-induced variation of the interfacial microenvironment, particularly the gas-liquid-solid triphase interface microenvironment, significantly influences the OER activity. Building on this theoretical insight, the authors propose a possible methodology to manipulate the local microenvironment to enhance OER by introducing hydrophobic molecules. The results are impressive, and the methods used are comprehensive. The conclusions are of interest to the photocatalysis community, and therefore, I recommend this manuscript for publication. However, there are a few comments listed below that are worth further clarification.

1. The Methods section mentions that the transition states (TS) were searched using a constrained optimization scheme. To ensure reproducibility, additional details, including the specific coordinate selected for the transition state search, should be provided.
2. The method for calculating the energy of steps involving holes needs clarification, particularly regarding how the authors handle the comparison of energies for species with different charges. The authors should provide a further explanation of the methodology used. For instance, in Supplementary

Table 2, the exothermicity of the third step by -0.20 eV at 298 K (l), corresponding to the energy difference between O^-H^+ and $\bullet\text{O}^-\text{H}^+$, should be thoroughly explained.

3. As the temperature increases, the interface environment undergoes a transition into a more disordered hydrogen bonding network. To provide a more precise analysis of the hydrogen bonds, it would be beneficial to compare the average number of hydrogen bonds per water molecule in the six cases described (Figure 1 a-f).

4. It seems to me that the charge recombination and the incoming light irradiation should be important for the OER rate since it affects the hole concentration directly. It would be of interest to have a more detailed description of how the holes concentration affects the OER activity.

5. This manuscript offers theoretical insights into the influence of the interface microenvironment on optimizing OER kinetics. It would be valuable to compare these theoretical findings with other experimental phenomena. For instance, the work of Lee et al. (Nat. Nanotechnol. 2023, 18, 754) demonstrated an enhanced water-splitting activity on floatable photocatalyst (Pt/TiO₂ cryoaerogel). Could this experimental result be considered in relation to the mechanistic proposal presented in this study?

Responses to Reviewer's Comments

Responses to Reviewer #1

The authors have fully addressed the concerns from the reviewers. I have no more comments. It can be accepted as is.

Response: We really appreciate the referee's recommendation for acceptance.

Responses to Reviewer #3

I appreciated the rebuttal from the Authors and all my questions have been answered, therefore I suggest to accept the manuscript.

Response: We really appreciate the referee's recommendation for acceptance.

Responses to Reviewer #4

Comments: *“This work presents a comprehensive computational investigation of the atomic-level processes of photocatalytic oxygen evolution reaction (OER) on the aqueous TiO₂(110) interface, utilizing first-principles density functional theory calculation, molecular dynamics simulation, and microkinetic analysis. The authors observed that the crucial intermediate OH[•] radicals involved in the photocatalytic OER exhibit semi-hydrophobic properties. They present quantitative evidence indicating that the temperature-induced variation of the interfacial microenvironment, particularly the gas-liquid-solid triphase interface microenvironment, significantly influences the OER activity. Building on this theoretical insight, the authors propose a possible methodology to manipulate the local microenvironment to enhance OER by introducing hydrophobic molecules. The results are impressive, and the methods used are comprehensive. The conclusions are of interest to the photocatalysis community, and therefore, I recommend this manuscript for publication. However, there are a few comments listed below that are worth further clarification.”*

Response: We really appreciate the referee's positive comments and valuable suggestions. The detailed responses to specific comments are given below.

Q1. “The Methods section mentions that the transition states (TS) were searched using a constrained optimization scheme. To ensure reproducibility, additional details, including the specific coordinate selected for the transition state search, should be provided.”

Response: We thank the reviewer for the valuable questions on the TS search method and the detailed descriptions have been added in the revised manuscript (line 318 on Page 15).

In this work, we used the constrained optimization scheme to capture the TS, developed in a previous study in our group (Phys. Rev. Lett. 1998, 80, 3650). As the schematic diagram illustrates in Figure R1, in the constrained minimization technique, the distance of the atoms that will form new bond is constrained at an estimated value and the total energy of the system is

minimized with respect to the other degrees of freedom. The TSs can be located via changing the fixed distance, and was verified when (i) all forces on atoms vanish (the criterion is set as 0.05 eV/\AA) and (ii) the total energy is a maximum along the reaction coordination but a minimum with respect to the rest of the degrees of freedom. For the relatively simple reaction (without the pericyclic reaction mode involved), this optimization method can achieve a similar reaction barrier faster than the general TS search approaches, such as NEB.

Taking the H_2O deprotonation at the gas condition as an example (i.e. $\text{H}_2\text{O}_{\text{ad}} + \text{O}_{\text{br}} \rightarrow \text{OH}_{\text{f}}^- + \text{O}_{\text{br}}\text{H}^-$), we can constrain the $\text{H}\cdots\text{O}_{\text{br}}$ distance as the reaction coordinate to search the TS, and the barrier from our constrained optimization method is 0.37 eV . Notably, this value is very close to the result calculated by the traditional CI-NEB method ($\sim 0.40 \text{ eV}$, see Figure R2 taken from *PNAS* **2017**, 114, 1801), indicating the accuracy of our TS search method. It is worth noting that this approach has been successfully applied in previous studies, such as photocatalytic OER, CH_4 activation, and CH_3OH oxidation, as well as some conventional catalytic reactions (*Nat. Catal.* **2018**, 1, 291; *J. Am. Chem. Soc.* **2023**, 145, 21897; *JACS Au* **2022**, 2, 188; *ACS Catal.* 2017, 7, 2374; *JACS Au* **2022**, 2, 2352).

Figure R1 Scheme of the constrained minimization technique to locate the transition state. The thick solid line is the minimal energy pathway from the initial state (IS) to the final state (FS). The transition state sits on top of the reaction pathway. The dotted line schematically represents the optimization procedure, in which the constrained minimization technique ‘forces’ the structure optimization ‘uphill’ towards the transition state.

Figure R2 Energy profiles of H_2O deprotonation on $\text{TiO}_2(110)$ via CI-NEB method and the corresponding optimized configurations of IS and FS (taken from *PNAS* **2017**, 114, 1801).

Q2. “The method for calculating the energy of steps involving holes needs clarification, particularly regarding how the authors handle the comparison of energies for species with different charges. The authors should provide a further explanation of the methodology used. For instance, in Supplementary Table 2, the exothermicity of the third step by -0.20 eV at 298 K (l), corresponding to the energy difference between OH_i^- and $\cdot\text{OH}_i$, should be thoroughly explained.”

Response: We thank the referee for this valuable comment, and the detailed descriptions have been added in the Method section in the revised manuscript (line 388 on Page 18).

The trapped holes or surface radicals were simulated by introducing an OH group on the opposite surface instead of extracting electrons from the system, in order to eliminate the possible energy error due to the background charge correction. After introducing an OH group as an electron acceptor on the bottom surface, an electron would be trapped at the OH group forming a surface OH_i^- , and a h^+ could thus be simulated in a charge neutral system. The localization of hole is confirmed by the electronic structure analysis, including the site-projected magnetic moment and Bader charge analysis. Such a method has been carefully tested in our previous work (*Nat. Catal.* **2018**, 1, 291; *Phys. Chem. Chem. Phys.* **2015**, 17, 1549; *JACS Au* **2022**, 2, 188). It is worth noting that, in this strategy, the uncertain and relatively small electron-hole interaction that varies with the trapping center in the TiO_2 system was neglected, while ensuring the electroneutrality of the periodic system.

Taking the third step in Supplementary Table 2 ($\text{OH}_i^- + \text{h}^+ \rightarrow \cdot\text{OH}_i$) as an example, we calculated the reaction energy by comparing the total energy of two systems: (i) a h^+ localized on a subsurface lattice oxygen with a OH_i^- on the surface; and (ii) a h^+ localized on the oxygen in the OH group (the $\cdot\text{OH}_i$ radical). Each system is charge neutral and therefore the total energy can be directly compared.

Q3. “As the temperature increases, the interface environment undergoes a transition into a more disordered hydrogen bonding network. To provide a more precise analysis of the hydrogen bonds, it would be beneficial to compare the average number of hydrogen bonds per water molecule in the six cases described (Figure 1 a-f).”

Response: We appreciate the referee for this valuable advice. We have added the hydrogen bond number/length analysis as requested in the revised Supplementary Information (Supplementary Fig. 1; Page S2).

In terms of the structural configuration, water molecules in the typical liquid phase interact with neighbouring molecules to form six-, five- or four-member ring structures. As the temperature increases, the interface environment undergoes a transition into a more disordered hydrogen bonding network. In the case of 500 K (coexist), the density of water molecules is already lower than that of the typical liquid water. In particular, under the condition depicted in Fig. 1f of the main text, the volume occupied by water molecules has almost doubled, resulting in a significant decrease in water density of 48%. Fig. R3 reveals the number and length of

hydrogen bonds as a function of the distance from the interface in six cases from the statistical analysis. As the volume increases, the average number of H-bonds per water molecule gradually decreases. Particularly, at the first and second water layers of the liquid/catalyst interface, there is a significant reduction in the number of H-bonds at 500 K (coexist). This demonstrates that the hydrogen bond network is to some extent destroyed once the liquid-gas transition begins to occur. Furthermore, as the H-bonds are disrupted, the average length of the hydrogen bonds also increases (Fig. R3b).

Fig. R3 Comparison of the distributions of H-bonds per water molecule along the interface. **a** Distribution of the average number of H-bonds per water molecule. **b** Distribution of the average length of H-bonds per water molecule.

Q4. “It seems to me that the charge recombination and the incoming light irradiation should be important for the OER rate since it affects the hole concentration directly. It would be of interest to have a more detailed description of how the holes concentration affects the OER activity”

Response: We thank the referee for this in-depth discussion and kind suggestion. We totally agree with the referee that the charge recombination and the intensity of light irradiation are very important for the overall photoactivity since they directly affect the hole concentration reaching the surface. As shown in Figure R4, the photoactivity of OER is strongly affected by the hole concentration reaching the surface. Specifically, we have investigated how the hole concentrations affect the OER rate under four typical conditions (298 K(l), 373 K(l), 500 K(l) and 500 K(coexist)) with the hole concentration ranging from 10^{-10} to 1 ML. The following can be observed: (i) The OER rates increase significantly from the very low concentration of the surface-reaching hole and reach a plateau at a certain threshold of the hole concentration. The threshold values become gradually smaller from 298 K(l)→373 K(l)→500 K(l)→500 K(coexist), indicating the relatively small dependence of the hole concentration on the liquid-vapor coexisting interface environments. (ii) Compared to the other three conditions, it consistently exhibits better OER activity under the 500 K (coexist) condition with varying hole concentration, aligning with the corresponding activity trend described in Fig. 1h of the main text.

Fig. R4 The logarithmic plots of the OER rates at different water/TiO₂(110) interfaces as a function of hole concentration.

Here we take the case of 500 K(coexist) as an example to further illustrate the effect of hole concentration. When the hole concentration is in the range from 10^{-10} to 10^{-4} ML, an increase in hole concentration leads to a higher rate of OER. When the hole concentration exceeds 10^{-4} ML, the OER rate reaches a plateau and the rate-determining step shifts, because at a higher hole concentration, the formation of $\cdot\text{OH}_i$ via trapping holes at the terminal OH_i^- becomes easier and is no longer the rate-determining step. Moreover, the presence of micro-bubbles under the liquid-vapor coexistence conditions slows down the water dissociation reaction and makes it earlier to be the rate-determining step (relative to the 298 K(l), 373 K(l) and 500 K(l) conditions). In this case, increasing the concentration of surface-reaching holes cannot further increase the reaction rate.

To take the points of the referee, the corresponding description has been added in revised Supplementary Information (Supplementary Fig. 4; Pages S6-S7).

Q5. “This manuscript offers theoretical insights into the influence of the interface microenvironment on optimizing OER kinetics. It would be valuable to compare these theoretical findings with other experimental phenomena. For instance, the work of Lee et al. (Nat. Nanotechnol. 2023, 18, 754) demonstrated an enhanced water-splitting activity on floatable photocatalyst (Pt/TiO₂ cryoaerogel). Could this experimental result be considered in relation to the mechanistic proposal presented in this study?”

Response: We thank the referee for the positive comment and for the valuable question. We have revealed that the $\cdot\text{OH}_i$ radical, as the pivotal intermediate of photocatalytic OER, has a unique relatively hydrophobic property, which results in the fact that in the process of $\cdot\text{OH}_i$ radical formation, it would consume additional energy to push away the water network. The elevated temperature influences the interfacial water distribution, making the hydrogen-bond network disordered and loose, which greatly facilitates the formation of semi-hydrophobic $\cdot\text{OH}_i$

radical. In addition, we have also shown that the liquid-vapor coexistence environment due to the temperature elevation can not only promote the $\cdot\text{OH}_t$ radical formation but also guarantee the function of the hydrogen-bonding water network for proton transfer and H_2O activation, thus achieving the superior OER performance. It is clear from our work that the microenvironment plays an important role in chemical reactions: The reaction rate can be increased by several orders of magnitude by adjusting the microenvironment; surprisingly, the microenvironment change due to temperature change is more important than the traditional temperature effect.

Our theoretical result could provide a mechanistic explanation for the very recent work showing that the floatable photocatalyst (Pt/TiO₂ cryoaerogel) can give enhanced water-splitting activity (*Nat. Nanotechnol.* 2023, 18, 754). As shown in **Figure R5**, compared with the the sunken nanocomposites, photocatalytic experiment using floatable nanocomposites proceeds at the air-water interface, which is more conducive to radical formation, such as $\cdot\text{OH}_t$ radical. Moreover, the high porosity of the nanocomposites can play a crucial role in providing easy access to water for the photocatalyst, which preserves the proton transfer channel for water deprotonation. Therefore, the floating photocatalyst achieves the high photocatalytic activity and deserves a further investigation and development.

Figure R5. Schematic of the photocatalytic experiment using floatable and sunken nanocomposites.

REVIEWERS' COMMENTS

Reviewer #4 (Remarks to the Author):

The submission has been greatly improved and is worthy of publication.